# Agreement between self-reported and researcher-measured height, weight and blood pressure measurements for online prescription of the combined oral contraceptive pill: an observational study

Hannah McCulloch [iD],[1] Alessandra Morelli [iD],[1] Caroline Free,[2] Jonathan Syred,[1] Riley Botelle [iD],[1] Paula Baraitser [iD][3,4]

For numbered affiliations see end of article.

**Correspondence to**
Hannah McCulloch;
hrmcculloch@gmail.com

## ABSTRACT

**Objective(s)** To compare agreement between self-reported height, weight and blood pressure measurements submitted to an online contraceptive service with researcher-measured values and document strategies used for self-reporting.

**Design** An observational study.

**Setting** An online sexual health service which provided the combined oral contraceptive pill, free of charge, to users in Southeast London, England.

**Participants** Between August 2017 and August 2019, 365 participants were recruited.

**Primary and secondary outcome measures** The primary outcome, for which the study was powered, was the agreement between self-reported and researcher-measured body mass index (BMI) and blood pressure measurements, compared using kappa coefficients. Secondary measures of agreement included sensitivity, specificity and Bland Altman plots. The study also describes strategies used for self-reporting and classifies their clinical appropriateness.

**Results** 327 participants fully described their process of blood pressure measurement with 296 (90.5%) classified as clinically appropriate. Agreement between self-reported and researcher-measured BMI was substantial (0.72 (95% CI 0.42 to 1.0)), but poor for blood pressure (0.06 (95% CI −0.11 to 0.23)). Self-reported height and weight readings identified 80.0% (95% CI 28.4 to 99.5) of individuals with a researcher-measured high BMI (≥than 35 kg/m$^2$) and 9.1% (95% CI 0.23 to 41.3) of participants with a researcher-measured high blood pressure (≥140/90 mm Hg).

**Conclusion** In this study, while self-reported BMI was found to have substantial agreement with researcher-measured BMI, self-reported blood pressure was shown to have poor agreement with researcher-measured blood pressure. This may be due to the inherent variability of blood pressure, overdiagnosis of hypertension by researchers due to 'white coat syndrome' or inaccurate self-reporting. Strategies to improve self-reporting of blood pressure for remote prescription of the combined pill are needed.

## STRENGTHS AND LIMITATIONS OF THIS STUDY

⇒ This is the first study to observe self-reporting of blood pressure and body mass index to obtain the combined oral contraceptive pill in a healthy population of reproductive age.

⇒ We compared blood pressure and body mass index measurements self-reported by users online with researcher-measured readings, taken on average 20 days later.

⇒ This study's assumption that researcher measurements reflect true blood pressure is problematic; however, clinicians taking single measures of blood pressure is standard practice in contraceptive provision.

⇒ We recognise the potential for self-selection bias in recruitment, where users who may have estimated their blood pressure during their order might have been unwilling to participate, and for healthy user effect since the majority of participants had previous experience of taking the pill.

## INTRODUCTION

In the United Kingdom (UK), 29.5% of sexually active women of reproductive age use oral contraceptives.[1] While contraception is available free of cost in the UK, research has found that 37.0% of women have difficulty accessing contraceptive services.[2]

Online sexual health services, offering tests for sexually transmitted infections ordered online and posted home, with self-taken samples posted to laboratories and results communicated remotely, have become part of the UK sexual health economy. Online services increase access to testing, including among those who have never tested before.[3 4]

Online services might facilitate access to contraception, enabling users to order

oral contraception with remote prescribing and postal delivery.[5] Remote prescription of the combined oral contraceptive (COC) requires a medical history and self-reported height, weight and blood pressure (BP). The accuracy of these biometric measurements is important to ensure appropriate, safe prescription.

The UK and WHO medical eligibility criteria (UKMEC/WHOMEC) classify all contraceptive contraindications on a scale of 1–4.[6 7] A classification of 1 or 2 implies that benefits of COC use outweigh the risks, while a classification of 3 or 4 suggests that risks of use outweigh benefits. A body mass index (BMI) equal to or greater than $35\,kg/m^2$ or a BP consistently above $140/90\,mm\,Hg$ would be classified as UKMEC 3 or 4.

We studied a UK-based, online contraceptive service, SH:24, which provided COC via post after an online medical history and self-reported height, weight and BP.[8] We aimed to compare self-reported BMI and BP measurements with measurements taken by researchers.

## MATERIALS AND METHODS
### Setting
SH:24 provided online sexual health services free to the residents of five areas of South East London, areas with poor sexual and reproductive health (SRH) indicators,[9 10] between January 2017 and August 2019. SH:24 is an NHS-commissioned service, registered with the UK Quality Care Commission and compliant with the General Medical Council guidance on remote prescribing.[11] SH:24 was advertised through online media, and signposting from clinics.

### Data collection
Users who ordered the COC were invited to a research visit within 18 weeks of ordering. At the visit, height and weight were measured. BP was measured with the participant sitting down having rested for 10 min, with the arm supported at heart level. Two cuff sizes were available, dependent on participant's arm circumference. Participants were asked to refrain from talking and crossing their legs during the measurements. Three readings were taken at 10-min intervals using a clinically validated, automated sphygmomanometer (Omron M6).[12] During this research visit, participants completed two surveys, documenting demographic information and strategies used to obtain their self-reported measurements.

### Data analysis
To compare self-reported with researcher-measured values, we report Kappa coefficients, as well as sensitivity, specificity, positive predictive value (PPV) and negative predictive value (NPV), of BP and BMI. The mean of the BP readings was used as the researcher-measured reading. Self-reported and researcher-measured BPs were classified as either UKMEC 1 and 2 or UKMEC 3 and 4. We have chosen to report sensitivity, specificity,

PPV and NPV of both BMI and BP, although we recognise that since data collection was completed, guidance has changed such that BP readings as measured by our researchers are no longer the 'gold standard' for diagnosing hypertension.[13] Limitations of using researcher measurements as the 'true' BP value are considered the discussion.

The primary outcome, for which the study was powered, was the kappa coefficient for self-reported versus researcher-measured BMI and BP. Three hundred and sixty-five participants were needed to achieve 90% power to detect a true kappa value in a test of a null hypothesis (H0): Kappa=0.7 versus an alternative hypothesis (H1): Kappa ≥0.9. This was based on a significance level of 0.05,[14] and a prevalence of UKMEC 3 and 4 BP and BMI of 7.5%, which was estimated from contraceptive consultations at a local sexual health service in one of the study areas of South East London.

We used paired t-tests to calculate mean differences between self-reported and researcher-measured weight, height and BMI, systolic and diastolic BP readings. We report Pearson's correlation coefficients for the level of correlation and present Bland-Altman plots with mean differences and 95% limits of agreement, defined as ±1.96× (SD of the difference). Analyses were completed using STATA (V.15) and Bland Altman plots were created using SPSS (V.24).

Participants were asked to describe how they obtained height, weight and BP measurements for their order. We classified these processes as clinically appropriate or inappropriate. Clinically appropriate was defined as a reading taken with a BP measurement device, or taken at a clinical location or pharmacy, or a manual reading taken by someone with the appropriate technical skill, within 12 months prior to ordering.

### Public and patient involvement (PPI)
Patients and the public participated in the identification of the importance of the research question for this study and the nested qualitative study.[15] Nationally significant PPI, endorsed by the UK Faculty of Sexual and Reproductive Health Care, found that among the general population 'research to evaluate which interventions increase uptake and continuation rates of effective contraceptive methods' was the top priority for UK academic contraceptive research.[16] Reflecting this, ahead of developing the study protocol, the research team conducted their own PPI prioritisation exercise with two focus groups of contraceptive service users, which identified research into online service delivery as a priority. The logic model for the intervention was similarly produced with involvement from patients and the public including interviews with 21 stakeholders.[5] Intervention development involved over 100 sexual health service users through a process of human-centred design.

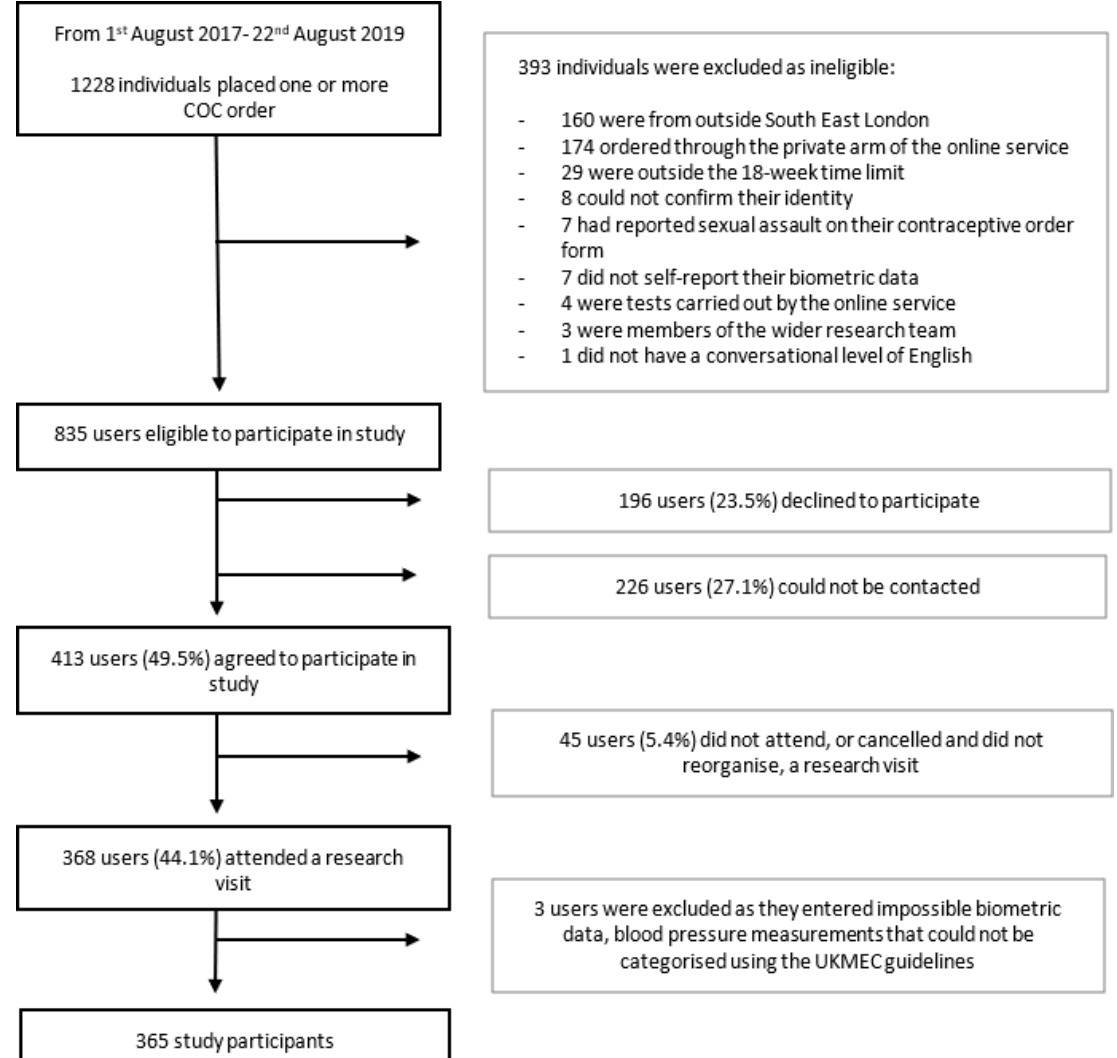

From 1st August 2017- 22nd August 2019

1228 individuals placed one or more COC order

393 individuals were excluded as ineligible:

- 160 were from outside South East London
- 174 ordered through the private arm of the online service
- 29 were outside the 18-week time limit
- 8 could not confirm their identity
- 7 had reported sexual assault on their contraceptive order form
- 7 did not self-report their biometric data
- 4 were tests carried out by the online service
- 3 were members of the wider research team
- 1 did not have a conversational level of English

835 users eligible to participate in study

196 users (23.5%) declined to participate

226 users (27.1%) could not be contacted

413 users (49.5%) agreed to participate in study

45 users (5.4%) did not attend, or cancelled and did not reorganise, a research visit

368 users (44.1%) attended a research visit

3 users were excluded as they entered impossible biometric data, blood pressure measurements that could not be categorised using the UKMEC guidelines

365 study participants

**Figure 1** Recruitment flow chart for study participants

## RESULTS

### Participants

Between August 2017 and August 2019, 1228 individuals ordered COC online, of which 835 were eligible for the study and 368 (44.1%) attended a research visit (figure 1). Median time between COC order and visit was 20 days (IQR: 14–30).

Participants had a mean age of 24 years, the majority (221/365, 60.6%) were White British and 86.6% (316/365) had higher educational qualifications. Almost all (351/365, 96.2%) had used COC before, 74.3% (271/365) had not ordered COC online before, and 87.9% (321/365) were prescribed the COC after a full assessment of medical eligibility. Eligible users who participated in the study were similar to those who did not although the proportion of new users was higher among non-participants (table 1).

### Strategies used to obtain measurements

The majority of participants (93.4%, 341/365) reported feeling confident self-reporting height, weight and BP, while 6.6% (24/365) did not. Of 365 participants, 327

provided data on their process of BP measurement that made classification of clinical appropriateness possible, and 90.5% of these (296/327) reported processes that were clinically appropriate (online supplemental material A).

Half of participants (50.1%, 181/361) measured their weight at home, 31.0% (112/361) reported a measurement from a clinic, 3.3% (12/361) from a pharmacy, 14.7% (53/361) from non-clinical setting, such workplace, gym or university, and 0.8% (3/361) participants reported estimating their weight measurements.

### Agreement between BP and BMI readings

Based on researcher-measured figures, the prevalence of UKMEC 3 and 4 BMI and BP was 1.4% (5/356) and 3.0% (11/365), respectively. Kappa statistics suggest that the agreement between self-reported and researcher-measured BMI data was substantial (0.72 (95% CI 0.42 to 1.00)), but agreement between self-reported and researcher-measured BP was poor (0.06 (95% CI –0.11 to 0.23)). Two-by-two tables from which agreement was derived are presented in online supplemental material B

**Table 1** Sociodemographic characteristics, contraceptive and online service use of population and sample

| | Eligible population n=835, %* | Non-participants n=470, % | Study participants n=365, % | |
|---|---|---|---|---|
| Sociodemographic characteristics | | | | |
| Age (years)* mean | 24.9 | 24.9 | 24.8 | 0.55† |
| 18–19 | 86 (10.3) | 57 (12.1) | 29 (7.9) | 0.05† |
| 20–24 | 360 (43.1) | 185 (39.3) | 175 (48.0) | |
| 25–34 | 355 (42.5) | 208 (44.3) | 147 (40.3) | |
| 35+ | 34 (4.1) | 20 (4.3) | 14 (8.8) | |
| Ethnicity* | | | | |
| White English/Welsh/ Scottish/Northern Irish/ British/ Other | 524 (62.7) | 303 (64.5) | 221 (60.6) | 0.10† |
| Black African/Caribbean/British/Other | 113 (13.5) | 56 (11.9) | 57 (15.6) | |
| Asian/Asian British | 86 (10.3) | 53 (11.3) | 33 (9.0) | |
| Mixed/multiple ethnic groups | 76 (9.1) | 34 (7.2) | 42 (11.5) | |
| Other ethnic groups | 31 (3.7) | 20 (4.3) | 10 (2.7) | |
| Not known/ prefer not to say | 6 (0.72) | 4 (0.85) | 2 (0.55) | |
| Index of multiple deprivation quintile*‡ | | | | |
| 1 (most deprived) | 282 (34.2) | 151 (32.7) | 131 (36.2) | 0.44† |
| 2 | 335 (40.7) | 187 (40.5) | 148 (40.9) | |
| 3 | 155 (18.8) | 89 (19.2) | 66 (18.2) | |
| 4 | 42 (5.1) | 29 (6.3) | 13 (3.6) | |
| 5 (least deprived) | 10 (1.2) | 6 (1.3) | 4 (1.1) | |
| Qualifications (attained or studying towards)§ | | | | |
| No academic qualifications | | | 1 (0.27) | |
| GCSES (or equivalent level) | | | 13 (3.6) | |
| AS/A Levels (or equivalent level) | | | 34 (9.3) | |
| Higher education qualifications (or equivalent level) | | | 316 (86.5) | |
| Not sure or other | | | 1 (0.27) | |
| Employment status§ | | | | |
| Employed | | | 265 (72.6) | |
| Parent/carer | | | 2 (0.6) | |
| Student | | | 93 (25.5) | |
| Unemployed | | | 5 (1.3) | |
| Contraceptive history and use | | | | |
| Used oral contraception before?§ | | | | |
| Yes | | | 352 (96.4) | |
| No | | | 13 (3.6) | |
| Taken combined oral contraceptive (COC) before?§ | | | | |
| Yes | 779 (93.3) | 428 (91.1) | 351 (96.2) | 0.003† |
| No | 56 (6.7) | 42 (8.9) | 14 (3.8) | |
| Reason for taking the pill?§ | | | | |
| To prevent pregnancy | | | 302 (82.7) | |
| For other reasons | | | 40 (11.0) | |
| To both prevent pregnancy and for other reasons | | | 23 (6.3) | |
| Online service use | | | | |

Continued

**Table 1** Continued

| | Eligible population n=835, %* | Non-participants n=470, % | Study participants n=365, % | |
|---|---|---|---|---|
| Ordered COC from online service before?* | | | | |
| Yes | | | 94 (25.7) | |
| No | | | 271 (74.3) | |
| Prescribed COC from online service? | | | | |
| Yes | 726 (86.9) | 405 (86.2) | 321 (87.9) | 0.45† |
| No | 109 (13.1) | 65 (13.8) | 44 (12.1) | |

*Reported by user at time of order with the online service.
†To compare demographic information, pill and service use between eligible users who did and did not participate in the study, t-test and $\chi^2$ test were carried out. P values are given to 2dp.
‡Index of multiple deprivation data unavailable for 11 users.
§Self-reported by participant at research visit, otherwise not collected by the online service.

and C, full results are presented in table 2. Self-reported height and weight readings identified 80.0% (95% CI 28.4 to 99.5) of individuals with a researcher-measured BMI corresponding to UKMEC 3 or 4. Self-reported BP readings identified 9.1% (95% CI 28.4 to 99.5) of those with a researcher-measured BP corresponding to UKMEC 3 or 4. The specificity of self-reported BMI and BP measurements was high at 99.4% (95% CI 98.0 to 99.9), and 97.2% (95% CI 94.9 to 98.6), respectively. Of the 10 participants who were found to have self-reported BP UKMEC 1 or 2, but measured as a 3 or 4, 8 reported clinically appropriate BP measurement processes and two reported unclassifiable processes.

Self-reported weight was on average 1.8 kg lower (95% CI −2.4 to −1.2) and the resulting BMI on average 0.70 kg/cm$^2$ lower (95% CI −0.94 to −0.47) than researcher-measured readings. Systolic BP was 8 mm Hg higher (95% CI 7.1 to 9.6) in the self-reported data than researcher-measured data. Average differences for height were <1 mm (95% CI 0.0002 to 0.008) and for diastolic BP were <1 mm Hg (95% CI −0.54 to 1.44) (see table 3).

Bland Altman plots (figures 2–4) for BMI and systolic and diastolic BP illustrate the degree of agreement between self-reported and researched-measured values.[17] Normal distribution of the differences was verified with a histogram. Limits of agreement are wider in systolic and diastolic BP than in BMI (also see table 3). The plots show that for both BP and BMI there was no trend in differences between measurements; participants with both a high and low mean BP or BMI from self-reported and researcher measurements were found to have variable differences in self-reported and measured BP or BMI.

## DISCUSSION
### Key findings
Self-reported BMI was found to have substantial agreement with researcher-measured BMI, unlike self-reported BP, which had poor agreement. The majority of participants undertook a clinically acceptable process in obtaining self-reported BP. On average, participants slightly under-reported their weight and over-reported their systolic BP. Bland Altman plots showed better agreement for BMI than for systolic and diastolic BP. The plots also showed no trends or systemic variation in differences between self-reported and researcher measurements.

### Where this sits within the literature
Research from Australia, Hong Kong, Finland, Malaysia, Sweden and the USA has found similar, substantial agreement for self-reported and researcher/clinician-measured weight and height for populations of all ages.[18–23] Similar to our findings, in other studies which compared both self-reported and researcher-measured BP and BMI, agreement was stronger between BMI measures than between BP measurements.[21 24] Studies comparing self-reported and researcher/clinician-measured BP have found varying levels of agreement, with higher agreement often found older populations.[24–26] A large cohort study (n=1537) in New Zealand found kappa between self-reported and measured BP in 18–44 year olds to be 0.26 (p<0.001), with a sensitivity of 18.1% and a specificity of 99.1%.[25] Within the same study, those aged 45–64 had an agreement of 0.49 (p<0.001), a sensitivity of 47.5%. A smaller study (n=200) of Italian women in their 50s found a kappa of 0.319 for self-reported compared with measured hypertension.[24] Self-measurement of BP is more accurate than clinician-measured BP for those who are hypertensive, at risk of hypertension or are pregnant, possibly because they receive support and training for correct measurement.[27–29] Self-monitoring of BP is becoming a common activity for some of these patient groups, for example, among both hypertensive and non-hypertensive pregnant women, and those who self-monitor may have better knowledge about their BP.[30]

Findings from a nested qualitative sample within this study reflect a difference in self-reporting BMI and BP experiences, where height and weight measurements were accessible and familiar but BP readings were not. The qualitative study showed that participants worked

**Table 2** Agreement between self-reported and researcher-measured values

| | N | Prevalence of UKMEC 3 and 4 (n, %) | Sensitivity % (95% CI) | Specificity % (95% CI) | Positive predictive value % (95% CI) | Negative predictive value % (95% CI) | Cohen's Kappa coefficient Kappa coefficient (95% CI) | P value |
|---|---|---|---|---|---|---|---|---|
| UKMEC categorisation for BMI (1 and 2 vs 3 and 4) | 365 | 5 (1.4%) | 80.0 (28.4 to 99.5) % | 99.4 (98.0 to 99.9) % | 66.7 (22.3 to 95.7) % | 99.7 (98.5 to 100.00) % | 0.72 (0.42 to 1.00) | 0.0000 |
| Blood pressure (<140/90 vs >or= 140/90) | 365 | 11 (3.0%) | 9.1 (0.23 to 41.3) % | 97.2 (94.9 to 98.6) % | 9.1 (0.23 to 41.3) % | 97.2 (94.9 to 98.6) % | 0.06 (−0.11 to 0.23) | 0.1156 |

hard to understand and measure BP through a combination of recent/past measurements, borrowed machines, health service visits and online research. This analysis suggests the importance of the acknowledgement of work required to measure BP, evidence of credible human support and a digital interface that communicates the health benefits of accurate measurement.[15]

### Strengths and limitations

This is the first study of its kind to observe self-reporting of BP and BMI to obtain the COC in a young, healthy population of reproductive age. It is timely as innovative, self-care interventions for SRH become more prevalent.[31 32]

We have compared self-reported and researcher-measured BP. Although, prior to contraception provision, clinicians taking single measures of BP is standard, the assumption that the measurement from the research visit reflects the true BP is problematic. BP is inherently variable, influenced by numerous extrinsic and intrinsic factors.[33] A systematic review of the sources of inaccuracy in the measurement of adult resting BP identified significant directional effects from 27 sources, resulting in a range of −23.6 to +33.0 mm Hg for systolic BP and −14.0 to +23.0 mm Hg for diastolic BP.[34] While researchers employed recommended strategies to mitigate inaccuracies in BP measurement, some factors remain irremediable, for example 'white coat syndrome', whereby researcher-, or clinician-, measured BP are artificially raised due to anxiety, or natural BP variation.[35 36] Recently updated UK guidance recommends ambulatory or home BP monitoring involving multiple repeated measures over a day for ambulatory or at least 4 days for home BP monitoring to diagnose hypertension.[13]

There was potential for self-selection bias during recruitment where users who may have estimated their BP during their order might have been unwilling to participate. To mitigate this, invitation communication, while providing sufficient information on the study procedure, did not emphasise validation of measurements (online supplemental material D). The study had good uptake, with 44.1% of eligible users attending a research visit, where researchers reiterated their independence from the online service.

Almost all study participants had experience of taking COC before ordering from the online service (96.2%, 351/365), a figure that is greater than among non-responders (91.1%, 428/470, p<0.01). This suggests potential healthy user bias, in that those with more experience of COC may be more likely to agree to participate. Such experienced COC users may be less likely to have high BP or BMI, and could be more familiar with, or more aware of the importance of, strategies to obtain biometric readings for remote prescription that might be clinically appropriate.

The medium time between COC order and research visit was 20 days. This may have impacted on participants' recall of the details of recent BP measurements. This is evident in the 38 unclassifiable participant processes: 31

**Table 3** Mean differences between self-reported and measured values; Pearson's correlation coefficient and Bland Altman limits of agreement

| | Mean (SD) | | | Pearson's correlation coefficient | Bland Altman limits of agreement |
|---|---|---|---|---|---|
| | Self-reported values | Measured values | Mean difference (95% CI); SD | | |
| Height (m) | 1.65 (0.07) | 1.65 (0.06) | 0.004 (0.000 to 0.008); 0.038 | 0.844 | – |
| Weight (kg) | 62.7 (10.4) | 64.5 (12.4) | −1.8 * (−2.4 to −1.2); 5.8 | 0.883 | – |
| Body mass index (kg/m$^2$) | 22.9 (3.6) | 23.6 (4.1) | −0.70 * (−0.94 to −0.47); 2.3 | 0.832 | - 5.2, 3.8 |
| Systolic blood pressure (mm Hg) | 115.8 (10.4) | 107.5 (9.7) | 8.36* (7.1 to 9.6); 11.9 | 0.300 | −15.4 to 32.1 |
| Diastolic blood pressure (mm Hg) | 74.8 (8.5) | 74.3 (7.4) | 0.45 (-0.54 to 1.4); 9.6 | 0.276 | −18.8, 19.7 |

*$P<0.05$, paired t-test, difference between self-reported and measured values versus Ho: mean(diff)=0.

participants gave the date of their most recent BP measurement between their order and visit, 7 were unclassifiable due to limited information in their response.

There are methodological limitations that may explain the low kappa coefficient. The kappa produced, in particular for BP, should be interpreted considering the method's paradoxes and problems. The two-by-two table for BP (online supplemental material C) produced, with it's perfectly symmetrically unbalanced marginal totals, satisfies both paradoxes outlined by Feinstein and Cicchetti, which, in addition to the low prevalence of researcher-measured high BMI and BP, perhaps provides an additional factor explaining the low kappa in light of high percentage agreement.[37]

### Meaning and mechanism

Explanations for our findings include the natural variability of BP, white coat syndrome, or inaccurate self-measurement or reporting.[35 36] Although our quantitative data do not allow us to explore whether our findings are due to the natural variability of BP or to inaccurate self-measurement or reporting, our qualitative research provides some insights.[15] Participants shared both experiences of white coat syndrome at previous consultations, and, in relation to their online COC order, difficulties obtaining a new BP measurement, remembering values from a recent reading, and difficulties interpreting or uploading readings, as well as a single participant who recounted misreporting despite a known high BP.

While BP limits of agreement are comparable to biologically plausible ranges established when estimating the impact of sources of inaccuracy,[34] the clinically relevant factor for remote COC prescription is not the magnitude of these limits, but whether they shift BP from a non-contraindication to a contraindication. Nevertheless, at

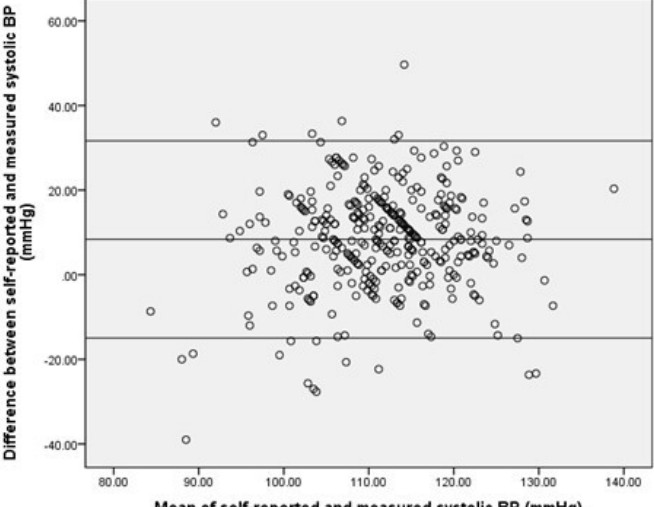

**Figure 2** Bland Altman plot of the differences between self-reported and researcher-measured body mass index (BMI) compared with the mean self-reported and researcher-measured BMI. The central line represents the mean difference between self-reported and researcher-measured BMI. The upper and lower boundaries represent the 95% limits of agreement.

**Figure 3** Bland Altman plot of the differences between self-reported and researcher-measured systolic blood pressure (BP) compared with the mean self-reported and researcher-measured systolic BP. The central line represents the mean difference between self-reported and researcher-measured systolic BP. The upper and lower boundaries represent the 95% limits of agreement.

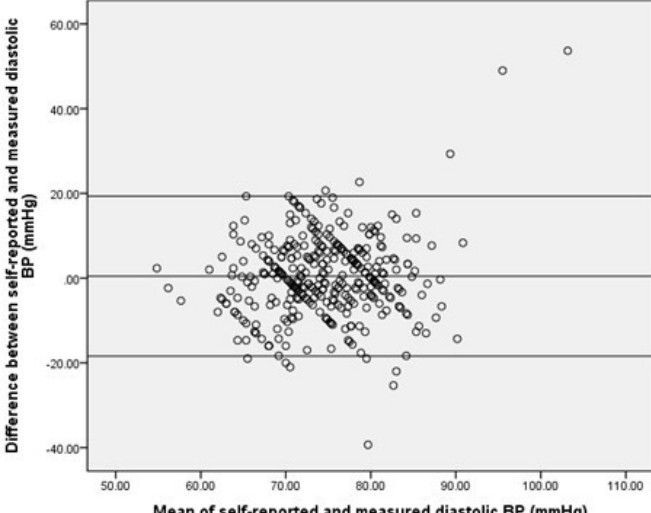

**Figure 4** Bland Altman plot of the differences between self-reported and researcher-measured diastolic blood pressure (BP) compared with the mean self-reported and researcher-measured diastolic BP. The central line represents the mean difference between self-reported and researcher-measured diastolic BP. The upper and lower boundaries represent the 95% limits of agreement.

their widest, such limits would trigger clinical concern and might initiate a conversation about process and accuracy of measurement.

The difference in findings for BP and BMI might reflect differences in familiarity with their measurement. Fifty years ago, accurate equipment for measuring weight were rare in households and most people did not know their current weight. BP monitors are now transitioning from a technology predominantly available in healthcare settings to one that is increasingly available at home. Strategies to increase accuracy of reporting could focus on improving understanding of BP and access to measurement.

## IMPLICATIONS AND RECOMMENDATIONS FOR ONLINE CONTRACEPTIVE SERVICES

This study's key findings, understood in the context of findings from the nested qualitative study,[15] have implications for both online and face-to-face contraceptive services. In order to maximise the benefits and minimise the risks of the online provision of COC, a self-care intervention, service providers need to take an active role in creating a 'safe and supportive enabling environment' for those seeking to obtain an online COC prescription. This includes acknowledging their responsibility in supporting users to self-report accurate biometric readings.[31 38]

In response to the findings, in collaboration with the study team, SH:24, the service evaluated in this study, redesigned their online COC clinical pathway. This was completed through a process of user-centred design. The design process suggested innovations that include: building skills for self-monitoring in face-to-face care when BP is measured by clinicians; providing home BP

monitors with support to use them; improved online information on the importance of BP measurements and consequences of inaccurate reporting; credible clinical support as required; issuing a short supply for the first COC order with information on how to obtain a BP measurement.[15 39] These strategies may improve the accuracy of BP reporting and we recommend further research to evaluate them.

## CONCLUSIONS

Online contraceptive services provide a convenient alternative for oral contraception users which may improve uptake and continuation.[5 40] Accuracy of self-reported height, weight and BP is important to ensure appropriate, safe COC prescription online, since BMI and BP may contraindicate use.[6 7]

While self-reported BMI was found to have substantial agreement with researcher-measured BMI, self-reported BP was shown to have poor agreement with researcher-measured BP. These differences may be due to the natural variability of BP, white coat syndrome, methodological limitations associated with kappa or by inaccurate self-measurement and reporting.

As online contraceptive services are increasingly prevalent and self-measured BP has been shown to be valid in other populations, strategies to improve BP self-measurement for remote prescription of COC are needed.

**Author affiliations**
[1]King's Centre for Global Health and Health Partnerships, King's College London Faculty of Life Sciences and Medicine, London, UK
[2]Department of Population Health, London School of Hygiene and Tropical Medicine Faculty of Epidemiology and Population Health, London, UK
[3]Sexual Health, King's College Hospital, London, UK
[4]Clinical and Evaluation, SH:24 CIC, London, UK

**Acknowledgements** We thank all users of the online service who took part in this study. We thank SH:24 for collaborating on the project, in particular the clinical team, designers and data analyst at the online service for their support with recruitment and data collection.

**Contributors** HM contributed to data collection, led the data analysis, and drafted the manuscript. AM contributed to data collection and data analysis and commented on the manuscript. CF and JS contributed to the conception and design of the study, contributed to data analysis and commented on the manuscript. RB contributed to data collection and commented on the manuscript. PB, Principal Investigator, responsible for the work and conduct of the study, led the conception and design of the study, oversaw data collection, contributed to data analysis and commented on the manuscript.

**Funding** This report is independent research funded by the National Institute for Health Research (Research for Patient Benefit Programme, PB-PG-0815-20009).

**Disclaimer** The views expressed in this publication are those of the author(s) and not necessarily those of the NIHR or the Department for Health and Social Care. The sponsor, King's College Hospital, was not involved in study design, data collection, analysis, or interpretation of data.

**Competing interests** Since completion of this study, HM has worked at the online service on a different research project. PB is clinical director of the online service studied and is one of the prescribers within this service. AM has also been employed by the online service as a bank clinical support midwife. This work is part of a process of research on innovation at SH:24 where all innovation is the subject of research to ensure that learning is shared widely. SH:24 is a 'not for profit' organisation that provides health services to the National Health Service and

sharing learning through innovation and research is one of the principles of the organisation.

**Patient and public involvement** Patients and/or the public were involved in the design, or conduct, or reporting, or dissemination plans of this research. Refer to the Methods section for further details.

**Patient consent for publication** Consent obtained directly from patient(s)

**Ethics approval** This study involves human participants and was approved by The East Midlands – Leicester Research Ethics Committee granted ethical approval to the project, 'Evaluation of online provision of oral contraceptives to measure: accuracy of self-reported height, weight and blood pressure; essential information transfer and user experience', on 6 June 2017. The reference for the ethical approval is 17/EM/0181. Participants gave informed consent to participate in the study before taking part.

**Provenance and peer review** Not commissioned; externally peer reviewed.

**Data availability statement** Data are available upon reasonable request. The anonymised dataset is available from the authors on request.

**ORCID iDs**
Hannah McCulloch http://orcid.org/0000-0001-5006-967X
Alessandra Morelli http://orcid.org/0000-0002-9803-2136
Riley Botelle http://orcid.org/0000-0002-3052-5698
Paula Baraitser http://orcid.org/0000-0002-3354-6494

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
