## [Reviewer comments · BMJ Open]

ARTICLE DETAILS

TITLE (PROVISIONAL)	Agreement between self-reported and researcher-measured height, weight and blood pressure measurements for online prescription of the combined oral contraceptive pill: an observational study
AUTHORS	McCulloch, Hannah; Morelli, Alessandra; Free, Caroline; Syred, Jonathan; Botelle, Riley; Baraitser, Paula

VERSION 1 – REVIEW

REVIEWER	Hannat Akintomide NHS Camden Provider Services, Sexual & Reproductive Health
REVIEW RETURNED	09-Nov-2021

GENERAL COMMENTS	Overall a well written paper relevant to contraceptive practice today. Explained analysis - agreement and/or correlation stats not popular in sexual health so good to see. Supplementary material provided for the more interested reader. Online prescribing is here to stay. Reassuringly and with measures to minimise self-selection bias, the study showed that majority (91%) accessing the online service had used the COC before and hence were familiar with the medication.
--

REVIEWER	Finn Egil Skjeldestad UiT Norges arktiske universitet, Department of Community Medicine
REVIEW RETURNED	19-Nov-2021

GENERAL COMMENTS	This is an important study as online prescriptions of hormonal contraception (HC) increases in several countries. In this respect it is crucial for the administrators to trust the information provided by the clients on BMI and BP. The study setting is appropriate for the study aims. However, for non-British readers, you need to spell out what SH:24 stands for. It is a clinical relevant approach to use WHOMEK/UKMEC criterias This is an important study as online prescriptions of hormonal contraception (HC) increases in several countries. In this respect, it is crucial for the administrators to trust the information provided by the clients on BMI and BP. The study setting is appropriate for the study aims. However, for non-British readers, you need to spell out what SH:24 stands for. It is a clinical relevant approach to use WHOMEK/UKMEC criteria's for contraindications for use of combined oral contraceptives (COCs). The authors state that "self-reported and researcher-measured BPs were classified as either UKMED 1&2 or UKMEC 3&4" (page 6, lines 135-36). How was this statement applied in the power calculations? What prevalence of UKMEC 3&4 was applied in these calculations? The authors need to present the prevalence of UKMEC 3&4 for both
---

	BMI and BP. It's seducing to present an 80% concordance of self-reported and researcher-reported high BMI when the prevalence of high BMI is 1.3 % (5:365), similar for high BP; prevalence 3.0; with a 9% (1:11) concordance. These issues need to be discussed. The abstract is too wordy with some unnecessary information. Conclusion too, long! The conclusion in the main text is too long. Write another paragraph on implications/recomentations of the study before the conclusion. In limitations I miss a paragraph on "healthy user effect" as the majority of participants have used COCs before study participation. Regarding references I miss this one: Self-Reported vs. Measured Height, Weight, and BMI in Young Adults. Olfert MD, Barr ML, Charlier CM, Famodu OA, Zhou W, Mathews AE, Byrd-Bredbenner C, Colby SE. Int J Environ Res Public Health. 2018 Oct 11;15(10):2216. doi: 10.3390/ijerph15102216. PMID: 3031426 I made a quick PubMed search - and there are more relevant literature.
--	---

VERSION 1 – AUTHOR RESPONSE

We thank the editor and both reviewers for their comments, which we have outlined and responded to, point by point, below.

- 1. Please revise the 'Strengths and limitations' section of your manuscript (after the abstract). This section should contain up to five short bullet points, no longer than one sentence each, that relate specifically to the methods.**

We have revised the strengths and limitations section of the manuscript (line 62-72), reducing the length of bullet points so that the overall word count of this section is similar to other recently published articles (for example, Jeffery, C., et al. Innovative approach to improve information accuracy in a two-district cross-sectional study in Bihar, India. *BMJ Open* 2022;12:e051427. doi:10.1136/bmjopen-2021-051427, and Schmidt CO., et al. Effects of whole-body MRI on outpatient health service costs: a general-population prospective cohort study in Mecklenburg-Vorpommern, Germany. *BMJ Open* 2022;12:e056572. doi:10.1136/ bmjopen-2021-056572). The revisions we have made aim to make the section more related to the methodology of the study, for example citing strengths such as the novel nature of the study, and limitations such as possible issues with researcher measurement as gold standard and potential biases.

- 2. For non-British readers, you need to spell out what SH:24 stands for**

SH:24 is the registered name of the non-for-profit organisation which delivers free online sexual health services across the UK, and in Ireland. It is not an abbreviation or acronym. We hope that the description in lines 111-115, as well as the reference which links them to the SH:24's website, will give readers a good understanding of the services provided by the NHS-commissioned provider.

- 3. It is a clinical relevant approach to use WHOMEK/UKMEC criteria for contraindications for use of combined oral contraceptives (COCs). The authors state that "self-reported and researcher-measured BPs were classified as either UKMED 1&2 or UKMEC 3&4" (page 6, lines 135-36). How was this statement applied in the power calculations? What prevalence of UKMEC 3&4 was applied in these calculations?**

We have added a sentence to state the prevalence of UKMEC 3&4 which was applied to the power calculations (line 137-139). The estimation for this prevalence was based of attendances of those seeking the combined contraceptive pill at a local sexual health clinic.

- 4. The authors need to present the prevalence of UKMEC 3&4 for both BMI and BP. It's seducing to present an 80% concordance of self-reported and researcher-reported high BMI when the prevalence of high BMI is 1.3 % (5:365), similar for high BP; prevalence 3.0; with a 9% (1:11) concordance. These issues need to be discussed.**

We have added an additional column to present the prevalence of UKMEC 3&4 for both BMI and BP into table 2. We have also added an additional sentence (line 198-199) reporting these findings in writing. The low prevalence of researcher-measured high BMI and BP has also been added as a methodological limitation to consider when interpreting the low kappa (line 299-300).

- 5. The abstract is too wordy with some unnecessary information. Conclusion too, long!**

We have redacted the abstract to make it shorter and more concise. We have removed additional information shared about median time between online order and research visit, where biometric readings were taken, which may not be necessary for the abstract. We have also shortened the conclusion in the abstract.

- 6. The conclusion in the main text is too long.**

We have shortened the conclusion in the main text.

- 7. Write another paragraph on implications/recommendations of the study before the conclusion.**

We have created a new subsection entitled "Implications and recommendations for online contraceptive services", before the conclusion (line 321-332). Similar to clinical implications sections we have seen in other BMJ Open papers, we have set out implications and recommendations for online contraceptive providers.

- 8. In limitations I miss a paragraph on "healthy user effect" as the majority of participants have used COCs before study participation.**

We have added a paragraph to the limitations section outlining the potential for health user affect amongst the sample (lines 288-293)

- 9. Re-review relevant literature on Pub- Med, updating references, including addition of suggested reference:**

- 1. Self-Reported vs. Measured Height, Weight, and BMI in Young Adults. Olfert MD, Barr ML, Charlier CM, Famodu OA, Zhou W, Mathews AE, Byrd-Bredbenner C, Colby SE. Int J Environ Res Public Health. 2018 Oct 11;15(10):2216. doi: 10.3390/ijerph15102216. PMID: 3031426**

We have returned to Pubmed and updated references, adding the reference suggested and other relevant papers, to include research into agreement between self-reported and measured height, weight and blood pressure from a broader range of countries (lines 240-244). We have also added recently published research into the prevalence of blood pressure self-monitoring amongst pregnant women (lines 253-155).

VERSION 2 – REVIEW

REVIEWER	Hannat Akintomide NHS Camden Provider Services, Sexual & Reproductive Health
REVIEW RETURNED	26-Jan-2022

GENERAL COMMENTS	Thank you for the opportunity to review this revised version of the manuscript. The paper has been improved and reads wells. I have no further comments.
--

REVIEWER	Finn Egil Skjeldestad UiT Norges arktiske universitet, Department of Community Medicine
REVIEW RETURNED	28-Jan-2022
GENERAL COMMENTS	The authors need to shorten and make the conclusion in abstract and at end of main text clearer. In addition, the "implications" section is poorly written, needs revision.

VERSION 2 – AUTHOR RESPONSE

We thank the editor and both reviewers for their comments. We have responded to the comments given by the second reviewer below.

Comment 1: “The authors need to shorten and make the conclusion in abstract and at end of main text clearer”

We have reworded the first sentence of the conclusion in abstract, and hope that this makes the section clearer (line 56-58). We feel that the three sentences in the abstract conclusion are all essential in conveying the paper’s conclusions, and would prefer to not shorten the conclusion anymore because of this.

We have reviewed recently published material from BMJ Open, and have found that the word count of conclusion in our abstract, and the conclusion in the main text, are in line with the length of corresponding sections in many of these articles.

Comment 2: “In addition, the "implications" section is poorly written, needs revision”

We have revised and rewritten several sentences in the implications section (lines 322 -330; line 334).